# A feedback loop that drives cell death and proliferation and its defect in intestinal stem cells

Shivakshi Sulekh[1,2], Yuko Ikegawa[1,3], Saki Naito[1,4], Asami Oji[5], Ichiro Hiratani[2,5], Sa Kan Yoo[1,2,6]

**Cell death and proliferation are at a glance dichotomic events, but occasionally coupled. Caspases, traditionally known to execute apoptosis, play non-apoptotic roles, but their exact mechanism remains elusive. Here, using *Drosophila* intestinal stem cells (ISCs), we discovered that activation of caspases induces massive cell proliferation rather than cell death. We elucidate that a positive feedback circuit exists between caspases and JNK, which can simultaneously drive cell proliferation and cell death. In ISCs, signalling from JNK to caspases is defective, which skews the balance towards proliferation. Mechanistically, two-tiered regulation of the DIAP1 inhibitor *rpr*, through its transcription and its protein localization, exists. This work provides a conceptual framework that explains how caspases perform apoptotic and non-apoptotic functions in vivo and how ISCs accomplish their resistance to cell death.**

## Introduction

Caspases are traditionally known to execute apoptosis (Kumar, 2007). Caspases induce cell death by cleaving important proteins such as the inhibitor of caspase-activated DNase (Enari et al, 1998). In general, strong activation of caspases leads to cell death, whereas weak or moderate activation can lead to a variety of phenotypes, including proliferation, differentiation, survival, and migration (Florentin & Arama, 2012; Nakajima & Kuranaga, 2017; Eskandari & Eaves, 2022). Some cells can also survive strong caspase activation, which is called anastasis (Tang et al, 2012; Sun et al, 2017, 2020). Although levels of caspase activation affect the resulting phenotypes (Florentin & Arama, 2012; Nano et al, 2023), how cells decide when to die or survive is enigmatic, especially when intermediate levels of caspase activation occur.

In the field of *Drosophila*, JNK is often described as a cell death inducer: JNK induces transcription of DIAP1 antagonists such as *rpr*/*hid*/*grim* (Luo et al, 2007; Shlevkov & Morata, 2012). However, the literature suggests that JNK also induces proliferation (Sun & Irvine, 2011; Enomoto et al, 2015; La Marca & Richardson, 2020). Although each molecular pathway that leads to cell death or proliferation has been clarified, what determines the outcome of JNK activation, either cell death or proliferation, has remained unclear.

In this report, we demonstrate that a positive feedback loop between JNK and caspases can drive proliferation and cell death simultaneously and that the pathway from JNK to caspases is defective in intestinal stem cells (ISCs), resulting in ISC proliferation upon caspase activation. This work provides a conceptual framework to interpret how caspase–JNK feedback simultaneously induces cell death and proliferation and the mechanism by which ISCs remain resistant to cell death.

## Results

ISCs are known to be difficult to kill. Genetic manipulation that induces apoptosis in other cell types does not kill them (Jiang et al, 2009; Lu & Li, 2015; Ma et al, 2016; Singh et al, 2016). Even complete starvation does not kill ISCs, which outlive the organism during starvation (Fig S1A and B). We recently found the first BH3-only protein, Sayonara (synr), in *Drosophila* (Ikegawa et al, 2023). Synr activates caspases through the Bcl-2 pathway, relatively moderately compared with Rpr-induced DIAP1 inhibition (Fig 1A) (Ikegawa et al, 2023). We found that mild activation of caspases by Synr, while inducing cell death in the wing disc, causes massive expansion of *esg*>GFP-labelled progenitor cells rather than cell death in the R4 region, where tissue turnover readily occurs (Ciesielski et al, 2022) (Fig 1B–D). Synr induces mitosis (Figs 1D and S1C), which occurs only in ISCs among cells in the gut (Miguel-Aliaga et al, 2018). In addition, the hyperplasia was observed when Synr was expressed specifically in ISCs (Fig S1D). Thus, although we cannot completely rule out an involvement of miss-differentiation in the expansion of *esg*>GFP cells, ISC proliferation contributes to the phenotype. Other components of the Bcl-2 pathway, such as *buffy*, *debcl*, and *dark*, also induce ISC proliferation and are necessary for Synr-induced ISC proliferation (Fig 1E and F). Caspase inhibition by microRNA for *rpr*/*hid*/*grim* (Siegrist et al, 2010) suppresses Synr-induced ISC

[1]Laboratory for Homeodynamics, RIKEN BDR, Kobe, Japan   [2]Division of Developmental Biology and Regenerative Medicine, Kobe University, Kobe, Japan   [3]Graduate School of Biostudies, Kyoto University, Kyoto, Japan   [4]Graduate School of Frontier Biosciences, Osaka University, Osaka, Japan   [5]Laboratory for Developmental Epigenetics, RIKEN BDR, Kobe, Japan   [6]Physiological Genetics Laboratory, RIKEN CPR, Kobe, Japan

Correspondence: sakan.yoo@riken.jp

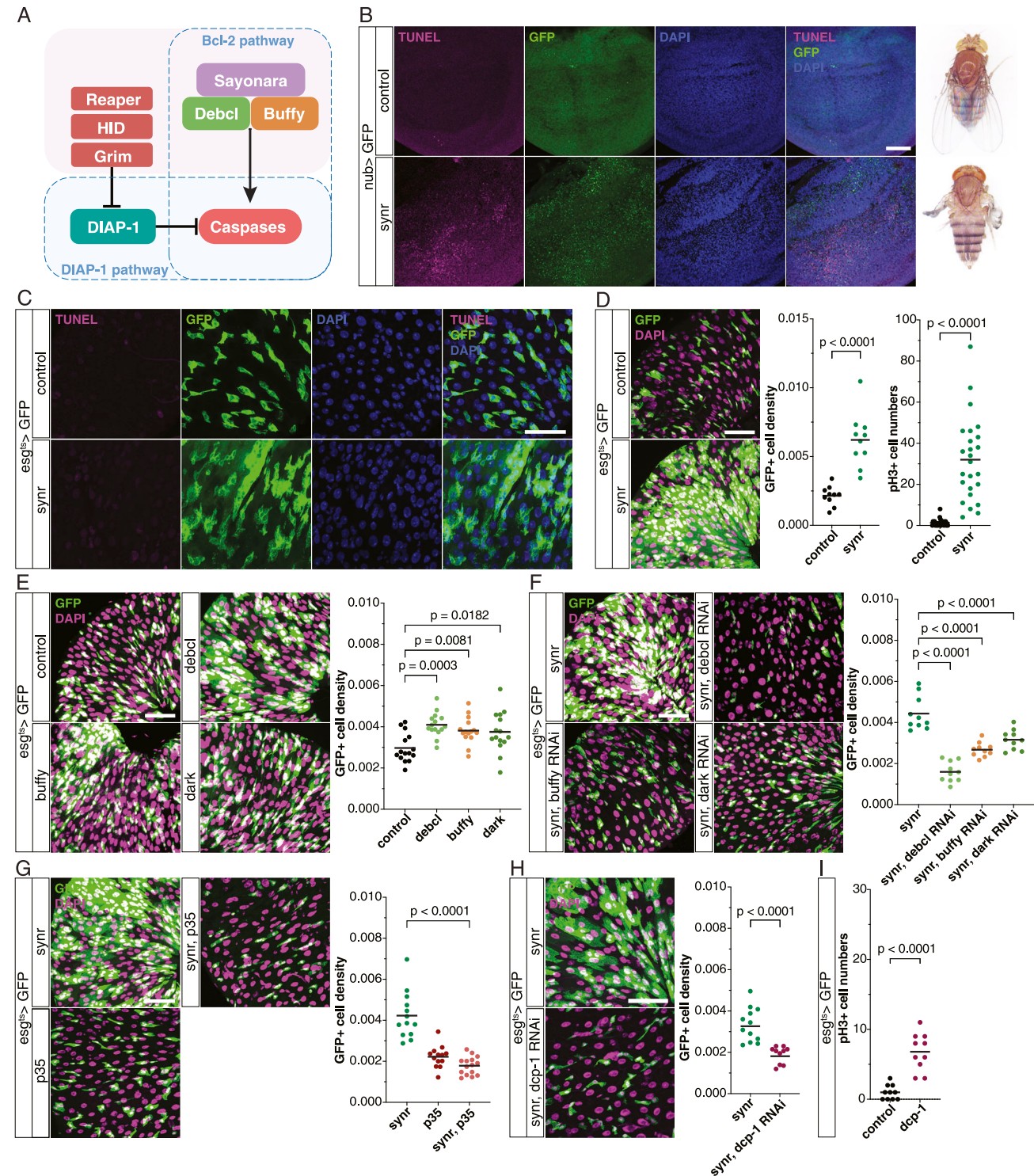

**Figure 1. Caspase-dependent proliferation of gut progenitors.**
**(A)** Schematic of the apoptosis pathway in *Drosophila*. **(B)** Synr in the wing pouch induces cell death, observed via TUNEL staining in the L3 wing disc and a structural defect in the adult wing. **(C)** Synr in gut progenitors does not induce cell death, observed via TUNEL staining in the R4 a-b region of the gut. **(D)** Synr induces proliferation of ISCs. Quantification is shown as GFP+ cell density (n = 10 each) and phospho-histone 3 (pH3)–stained mitotic cells (n = 25 each). **(E)** Bcl-2 pathway components, Debcl, Buffy, and Dark, induce proliferation of gut progenitors (n = 15 each). **(F)** Synr-induced proliferation is suppressed by knockdown of Debcl, Buffy, or Dark (n = 10 each). **(G)** Synr-induced proliferation is suppressed by p35 expression (n = 13, 13, 15). **(H)** Synr-induced proliferation is suppressed by knockdown of Dcp-1 (n = 12, 10). **(I)** Dcp-1 in gut progenitors for 1 d induces cell proliferation, which was detected by pH3 staining (n = 10 each). (D, E, F, G, H, I) Data information: statistical significance was determined using a two-tailed unpaired *t* test (D, G, H, I), and one-way ANOVA with Dunnett's post hoc test (E, F). Scale bars, 50 μm.

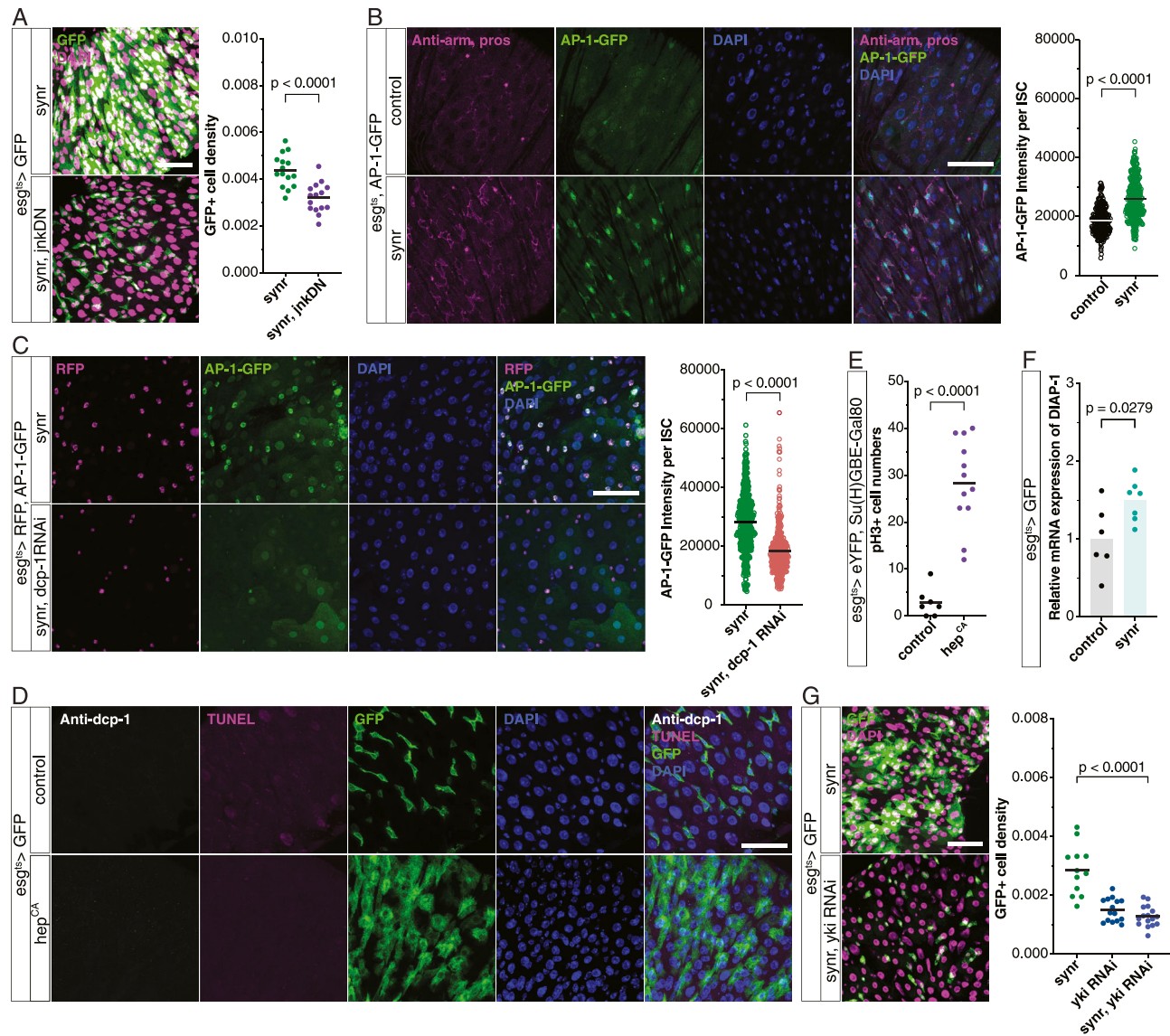

**Figure 2. JNK functions downstream of Synr/caspase signalling and induces proliferation via Yki.**
**(A)** Synr-induced proliferation is suppressed by the expression of the dominant negative form of JNK (n = 15 each). **(B)** Synr induces activation of JNK signalling, which was detected by the AP-1-GFP reporter. Synr was expressed in *esg+* cells for 2 d. ISCs and EEs were observed with anti-armadillo and anti-Prospero staining (armadillo-stained junctions mark ISCs and the other smaller cells, and EE cells identified by the Prospero marker were not considered for quantification). Quantification of AP-1-GFP intensity is shown (n = 8 guts, 265 cells; and 8 guts, 405 cells). **(C)** Synr-induced JNK activation is suppressed by Dcp-1 knockdown (n = 7 guts, 428 cells; and 8 guts, 280 cells). **(D)** Expression of the constitutively active form of hep (hep^CA) in gut progenitors for 1 d does not induce cell death, as observed by cleaved Dcp-1 and TUNEL staining. **(E)** Hep^CA expression specifically in ISCs for 1 d induces proliferation (n = 7, 12). **(F)** Synr induces the mRNA expression of a Yki target, *diap1*, in the midgut (n = 6, 7). **(G)** Knockdown of *yki* suppresses Synr-induced proliferation (n = 12, 15, 16). Data information: statistical significance was determined using a two-tailed unpaired *t* test. Scale bars, 50 μm.

proliferation (Fig S1E). The expression of p35 suppresses Synr-induced ISC proliferation, indicating that the phenotype depends on executioner caspases (Fig 1G). Specifically, the executioner caspase Dcp-1, which also works downstream of Synr in wing discs (Ikegawa et al, 2023), was necessary for the phenotype (Fig 1H). Dcp-1 expression for a short period was sufficient to induce ISC proliferation (Fig 1I), whereas its longer expression ablated ISCs (Fig S1F).

What mediates caspase-induced ISC overproliferation? Among pathways that we investigated, we found that JNK is a key: JNK

inhibition suppresses Synr-induced ISC proliferation (Fig 2A). Synr activates JNK signalling, which was detected by the AP-1 reporter (Chatterjee & Bohmann, 2012), through Dcp-1 (Fig 2B and C). JNK activation is sufficient to induce ISC proliferation without activation of Dcp-1 (Fig 2D and E), which is consistent with JNK's role in ISC proliferation (Biteau et al, 2008). Altogether, these data indicate that Synr/caspase signalling induces cell proliferation through JNK. Literature suggests that JNK can induce proliferation through Yki both autonomously and non-autonomously (Sun & Irvine, 2011; La Marca & Richardson, 2020). Indeed, we found that Synr induces

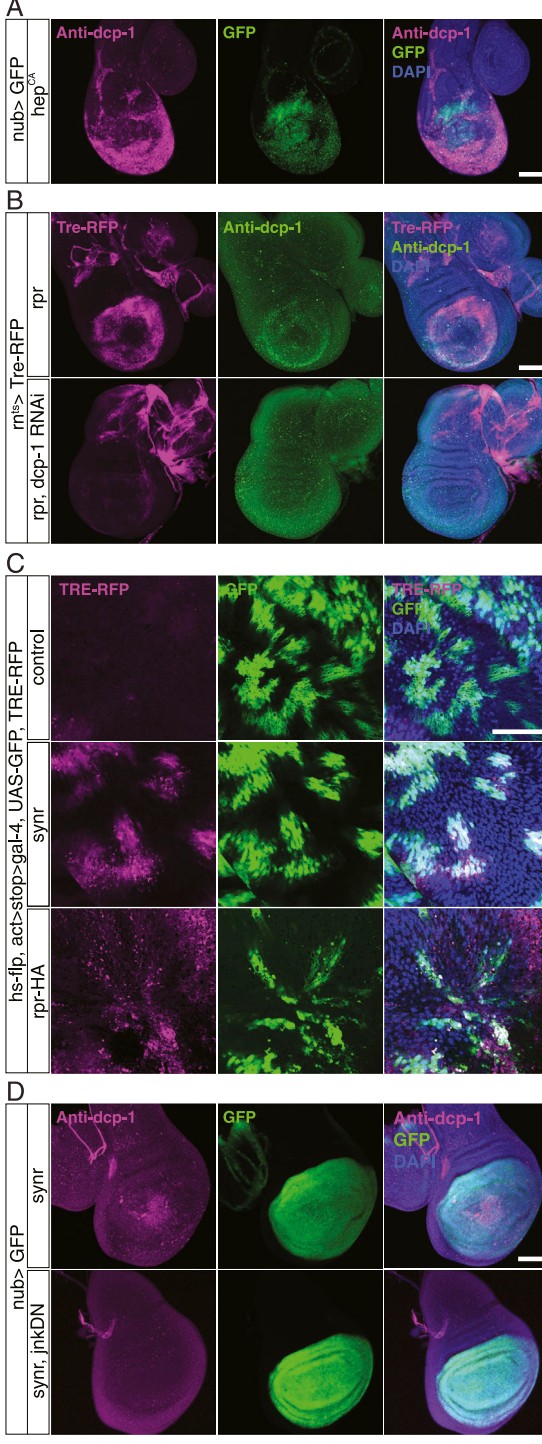

**Figure 3. JNK/caspase form a positive feedback circuit.**
**(A)** Expression of hep[CA] in the wing pouch of L3 larvae induces caspase activation, detected by anti-Dcp-1 staining. For unknown reasons, JNK-mediated caspase activation is more pronounced in the ventral periphery of the wing pouch. **(B)** Rpr-induced JNK (observed by Tre-RFP) and caspase activation is suppressed by Dcp-1 knockdown. **(C)** L3 wing disc bearing hs-flp–induced mosaics of WT or GFP+ clones, expressing synr/rpr along with Tre-RFP. **(D)** Synr-induced caspase activation in the wing disc is suppressed by the expression of the dominant negative form of JNK. Data information: statistical significance was determined using a two-tailed unpaired *t* test. Scale bars, 100 *μm* (A, B, D) and 50 *μm* (C).

transcription of a Yki target DIAP-1 (Fig 2F) and that Yki inhibition suppresses Synr-induced ISC proliferation (Fig 2G).

Thus far, we elucidated that caspases induce proliferation through JNK/Yki signalling in ISCs. As previously mentioned, there is accruing evidence that signalling from JNK to caspases through *rpr/hid* induces caspase activation (Luo et al, 2007; Shlevkov & Morata, 2012). Most of the previous studies were performed in imaginal discs. In addition to the JNK/caspase signalling, the caspase–JNK pathway, which we showed above, has been previously described, albeit only in the context of cell death (Shlevkov & Morata, 2012). The positive feedback between JNK and caspase was suggested to function as a powerful feedback loop that ensures cell death in the wing disc (Shlevkov & Morata, 2012). We also validated the JNK/caspase feedback in the wing disc by confirming that JNK activates caspase, that caspase activation through either Synr or Rpr activates JNK, and that inhibition of JNK suppresses Synr-induced caspase activation (Fig 3A–D). In contrast to the previous model that solely focused on cell death, because JNK clearly induces proliferation through Yki, we propose that this feedback between JNK and caspase has a potential to activate cell death and proliferation simultaneously. In our model, which outcome, death or proliferation, results from the feedback circuit depends on a balance of signals. Clearly, the balance is skewed towards cell proliferation in ISCs.

We hypothesized that if the feedback from JNK to caspases, which should operate independently from JNK-Yki–induced proliferation, is defective, it could weaken caspase-induced cell death, potentially favouring proliferation more. First, we tested whether Synr can induce transcription of *rpr*, which mediates JNK-induced caspase activation, in ISCs by performing qRT-PCR. We found that Synr cannot induce *rpr* efficiently in ISCs, in contrast to its induction in wing discs (Fig 4A and B). We also clarified that Synr or JNK activation cannot induce *rpr* expression in ISCs using rpr-lacZ (Fig 4C and D). Because Synr activates JNK itself (Fig 2B), we postulated that the chromatin status around *rpr* might be closed in ISCs. To test this hypothesis, we collected *esg*-GFP cells by FACS and performed ATAC-seq, a method to identify open chromatin regions. ATAC-seq demonstrated that the chromatin structure around *rpr* is closed (Fig 4E), which is in clear contrast to the open chromatin of *rpr* in the wing discs (Fig S2), which was analysed from the previous literature (Vizcaya-Molina et al, 2018). We also noted that the previous literature had suggested that ectopic Rpr cannot induce apoptosis in ISCs (Jiang et al, 2009; Lu & Li, 2015; Ma et al, 2016; Singh et al, 2016) and confirmed it ourselves: Rpr, which readily causes apoptosis in the wing disc, induces no effect or a very mild effect on ISCs, never leading to their complete depletion (Figs 5A and B and S3A).

These data indicate that there is a two-tiered mechanism that inhibits *rpr* in ISCs: *rpr* induction and its protein function. This can be considered as a fail-safe mechanism. We further pursued why overexpressed Rpr, which is usually very potent to induce cell death in other tissues (Goyal et al, 2000; Shlevkov & Morata, 2012), was not functional in ISCs. Rpr becomes functional when it localizes to mitochondria (Sandu et al, 2010). We found that Rpr is not specifically localized to mitochondria in ISCs (Fig 5C), which is contrasting to its localization in mitochondria of the wing disc cells (Sandu et al, 2010), suggesting a possibility that ectopic Rpr cannot induce cell death efficiently because of its failed localization to mitochondria. To test this idea, we forced localization of Rpr to

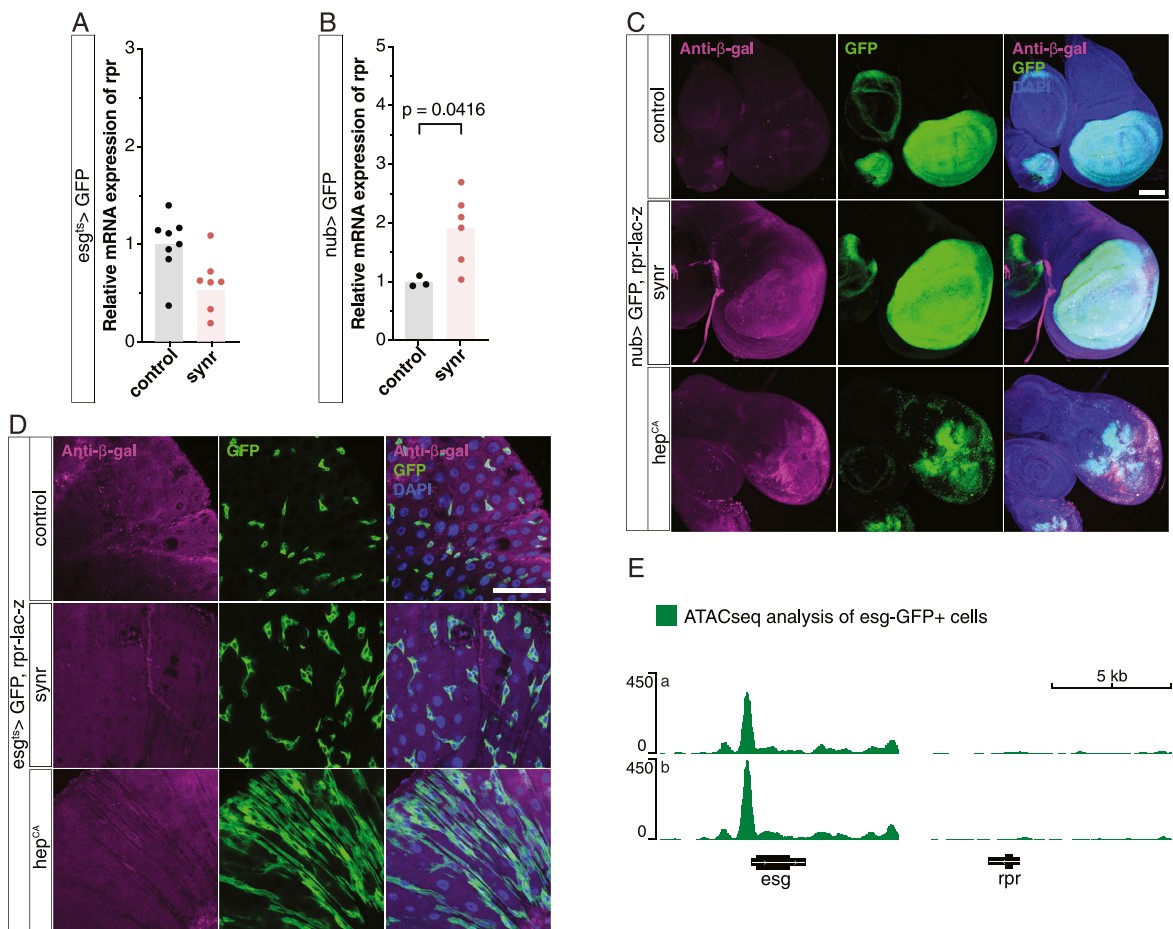

**Figure 4. JNK/caspase feedback circuit is defective in ISCs, because of the rpr chromatin state.**
**(A)** Synr does not induce the mRNA expression of *rpr* in the midgut (n = 8 each). **(B)** Synr induces the mRNA expression of *rpr* in the wing disc of L3 larvae (n = 3, 6). **(C)** Synr and Hep^CA induce rpr-lac-z expression in the wing disc. **(D)** Synr and Hep^CA do not induce rpr-lac-z expression in the gut progenitors. **(E)** ATAC-seq data indicate that the chromatin structure around *rpr* is closed, which is in contrast to the open chromatin around *esg*. Two replicates (a, b) are shown. **(C, D)** Data information: statistical significance was determined using a two-tailed unpaired *t* test. Scale bars, 100 *μ*m (C) and 50 *μ*m (D).

mitochondria by expressing *rpr* that has a mitochondrion locali-zation sequence (Sandu et al, 2010). Forced localization of Rpr to mitochondria depleted ISCs completely (Fig 5B and D) in the whole gut, except the R3 region (Fig S3B). To exclude a possibility that the difference between Rpr WT and Rpr-MTS is due to some genetic backgrounds, we newly generated UAS-Rpr WT and UAS-Rpr-MTS by inserting each on the same landing site (attP2). In this setting, Rpr-MTS (attP2) was more potent than Rpr WT (attP2) (Fig S3C). Fur-thermore, we also compared Rpr targeted to either mitochondria, ER, or Golgi and found that its targeting only to mitochondria ablates ISCs (Fig 5D). To localize WT Rpr to mitochondria using a different approach, we combined Rpr and the weak expression of Hid, which can help localize Rpr to mitochondria (Sandu et al, 2010). The low expression of Hid itself did not kill ISCs, but its combination with Rpr killed ISCs (Fig 5E), supporting the idea that localization of Rpr to mitochondria is defective in ISCs.

Our data indicate that caspase signalling leads to proliferation in ISCs because the feedback from JNK to caspase is defective. If this is the case, we should be able to induce massive proliferation in other tissues by creating a situation like ISCs. As a proof of principle, we mimicked the situation of ISCs by expressing microRNA for *rpr/hid/grim* (Siegrist et al, 2010) in wing disc cells. This arrangement should inhibit the feedback from JNK to caspase, similar to the status of ISCs. We found that a combination of JNK activation and the miRNA leads to cell proliferation (Fig 6A), validating our hypothesis based on the feedback loop model.

## Discussion

Here, taking advantage of the phenomenon that moderate acti-vation of caspase leads to ISC proliferation, we demonstrate the existence of the positive feedback loop between JNK and caspases, which can promote both cell death and proliferation. This modifies the previous feedback loop that was suggested to promote only cell death (Shlevkov & Morata, 2012). In ISCs, JNK-mediated caspase activation is defective because of the two-tiered fail-safe mech-anism, leading to cell proliferation.

We propose that this feedback between JNK and caspases could be a general mechanism that regulates cell death and proliferation.

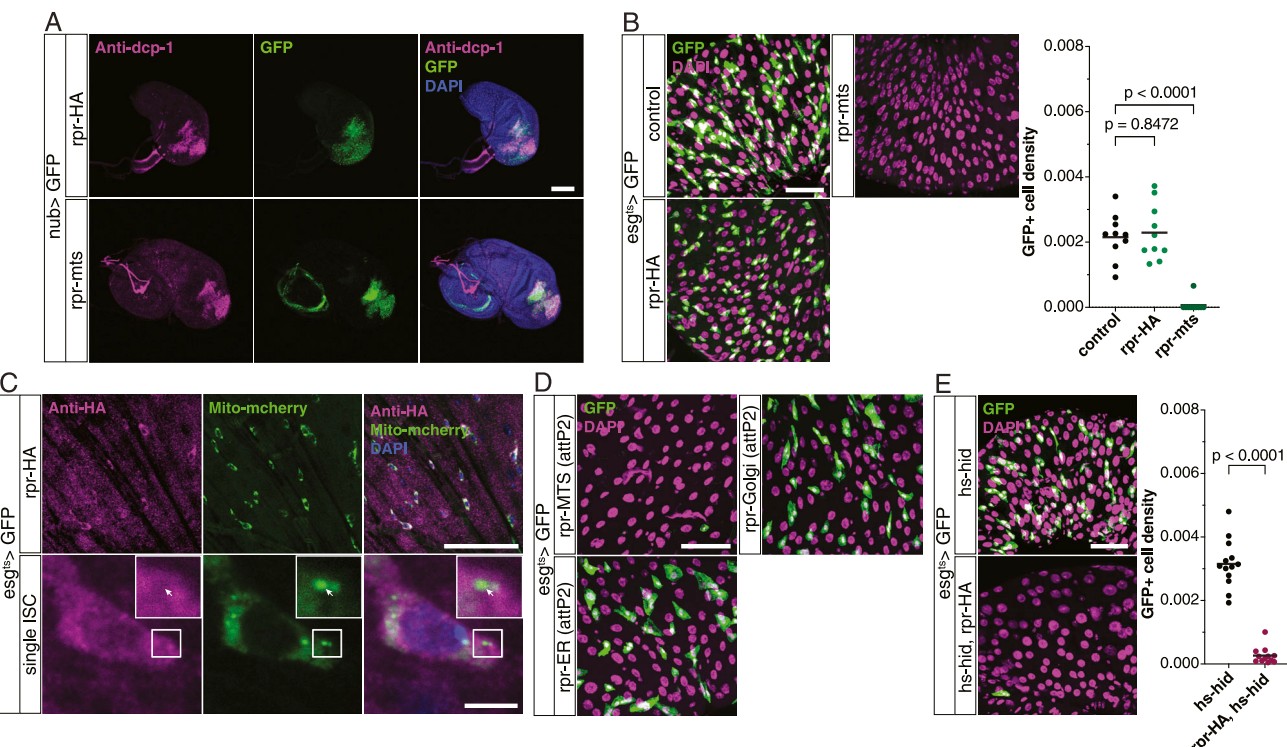

**Figure 5. Two-tiered regulation of Rpr in ISCs.**
**(A)** Rpr-HA or Rpr-mts expression in the wing pouch induces caspase activation. **(B)** Rpr-HA does not ablate gut progenitors, but Rpr-mts does (n = 10 each). **(C)** Rpr-HA (stained using anti-HA) does not colocalize with mito-mCherry. **(D)** Flies constructed using the same attP2 landing sites also show Rpr-MTS (attP2) as a strong ablator, as compared to Rpr-ER (attP2) and Rpr-Golgi (attP2). **(E)** Combination of Rpr-HA and hs-hid ablates gut progenitors (n = 13, 12). **(B, D)** Data information: statistical significance was determined using one-way ANOVA with Dunnett's post hoc test in (B) and two-tailed unpaired $t$ test in (D). **(A, B, C, D, E)** Scale bars, 100 $\mu$m (A), 50 $\mu$m (B, D, E), and 50, 5 $\mu$m (C).

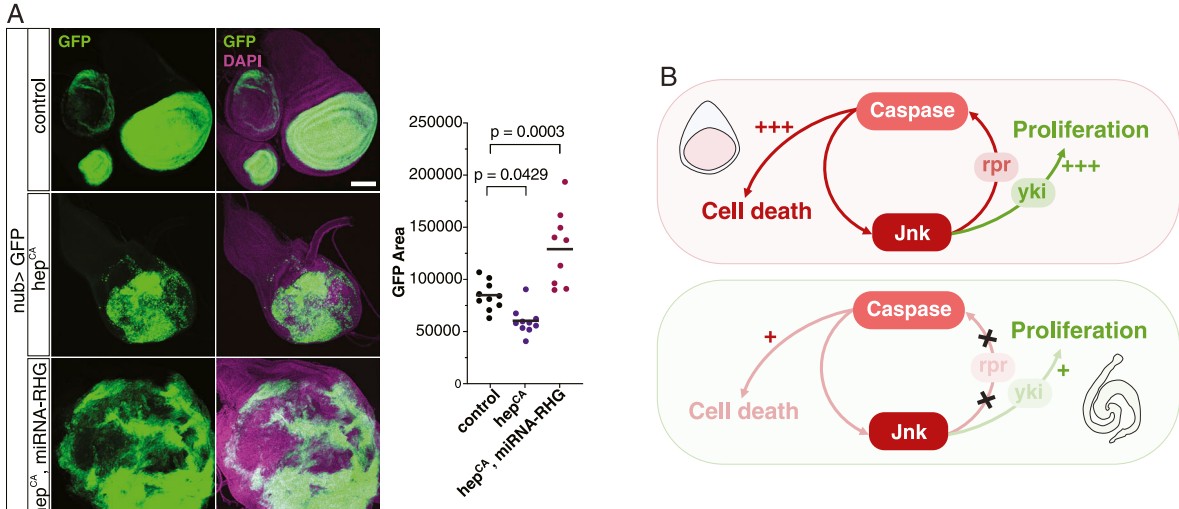

**Figure 6. Mimicking Rpr inefficiency in wing discs leads to tissue overgrowth.**
**(A)** Combination of Hep$^{CA}$ and miRNA for rpr/hid/grim in the wing pouch of L3 larvae leads to overgrowth of GFP+ cells (n = 10, 10, 9) **(B)** Schematic of the feedback circuit between JNK and caspase leading to cell death and proliferation. In the gut/ISCs, feedback from JNK to caspase is defective, which weakens signalling to both cell death and cell proliferation. Because cell proliferation is assumed to be more sensitive to weaker signals from the circuit, cell proliferation still occurs in ISCs. Similarly, the feedback generates both stronger cell death and proliferation in wing discs, but as the cell gets removed from the tissue upon apoptosis, proliferation is not observed. Data information: statistical significance was determined using one-way ANOVA with Dunnett's post hoc test. Scale bar, 100 $\mu$m.

The resulting outcome depends on the balance of this feedback. We note that if the feedback mechanism from JNK to caspase is defective as in ISCs, not only cell death signals but also proliferation signals should be suppressed (Fig 6B). Thus, for this model to be consistent with our observation, proliferation needs to be more sensitive to the feedback circuit than cell death. This is consistent with the idea that moderate caspase activation induces a variety of non-lethal phenotypes, whereas high caspase activity induces apoptosis (Florentin & Arama, 2012; Nakajima & Kuranaga, 2017; Eskandari & Eaves, 2022). We also found that Dcp-1 activation initially induces proliferation in ISCs (Fig 1I), which is followed by cell death if Dcp-1 is expressed for a longer time (Fig S1F). This is consistent with the previous study that implicated Dcp-1 in ISC death induced by inhibition of lipolysis (Aggarwal et al, 2022). We reason that Dcp-1 does not require the feedback mechanism to induce apoptosis when it is highly expressed. In general, if cell death signalling is strong enough, that is, if activated caspases cleave many important substrates, cells should die no matter how strong the proliferation signal is.

Previously, it was noted that JNK activation or its upstream activator Eiger/TNF induces either cell death or proliferation. Although Ras was suggested to affect JNK signalling during Ras-mediated transformation, there has been no conceptual framework to interpret seemingly contradictory phenotypes under the physiological condition (Enomoto et al, 2015; La Marca & Richardson, 2020). We provide a conceptual framework to interpret these dichotomous observations.

JNK is known to induce cell proliferation both autonomously and non-autonomously (Biteau et al, 2008; Sun & Irvine, 2011; Enomoto et al, 2015; La Marca & Richardson, 2020). Although we cannot rule out a possibility that JNK-Yki–mediated ISC proliferation involves non–cell-autonomous signals, we propose that the feedback loop likely occurs cell-autonomously in ISCs because ISCs are sparsely located, we manipulate genes autonomously, and multiple reports previously demonstrated JNK induces ISC proliferation autonomously (Biteau et al, 2008; Buchon et al, 2009; Biteau & Jasper, 2011; Meng & Biteau, 2015; Loudhaief et al, 2017; Hu & Jasper, 2019; Mundorf et al, 2019; Herrera & Bach, 2021). Thus, most likely this is different from the non-autonomous compensatory proliferation or apoptosis-induced proliferation, which is driven by cell death of surrounding cells (Perez-Garijo et al, 2004; Ryoo et al, 2004; Kondo et al, 2006; Fan & Bergmann, 2008; Mollereau et al, 2013).

Physiologically, why does the feedback system that simultaneously promotes death and proliferation exist? Previously, we demonstrated that oncogenic signalling that induces proliferation often causes cell death (Nishida et al, 2021). A similar concept could be applied here. We consider that the ancestral role of caspases is in cell proliferation, which is supported by literature (Shinoda et al, 2019; Eskandari & Eaves, 2022). This also intuitively makes sense: the scenario that a mechanism of cell proliferation acquired cell death seems more probable than the other way around. Thus, we reason that even in the modern time, caspases could play the ancestral role in regulating cell proliferation. To support this idea, the endogenous expression of Dcp-1 was shown to be important for imaginal disc growth previously (Shinoda et al, 2019).

Regarding ISC resistance to cell death, we speculate that such a system exists because tissue stem cells are important for organisms and should not die easily. Among ISCs, it is interesting to note that ISCs in the R3, gastric stem cells in the copper region, are

the most resistant to apoptosis. We speculate that the harsh acidic environment of the R3 (Strand & Micchelli, 2011) might make gastric stem cells especially resistant to cell death. Resistance to cell death can be accomplished through a variety of mechanisms. In ISCs, suppression of *rpr* is mediated by the two-tiered fail-safe mechanism at the chromatin and protein levels. The epigenetic regulation of *rpr* is similar to its regulation in the salivary gland (Zhang et al, 2014), and distinct from apoptosis resistance in cell cycle–arrested cells (Ruiz-Losada et al, 2022). The role that caspases play in ISCs is distinct from their apoptotic or non-apoptotic roles in enteroblasts (Reiff et al, 2019; Arthurton et al, 2020; Lindblad et al, 2021). Because the ancestral status such as the preservation of genes that predate animal origin is a key feature of tissue stem cells (Alie et al, 2015), we speculate that ISCs may accomplish their relative immortality by keeping the ancestral role of caspases.

# Materials and Methods

### *Drosophila* husbandry

Flies were maintained as previously described (Sasaki et al, 2021). The fly food was composed of 0.8% agar, 10% glucose, 4.5% corn flour, 3.72% dry yeast, 0.4% propionic acid, and 0.3% butyl p-hydroxybenzoate. For experiments with Gal80$^{ts}$, flies were raised at 18°C, virgin females were collected for three consecutive days, and the temperature was shifted to 30°C on the fifth day. Flies were flipped to a new food vial every 2 d. Genes were induced for 7 d, unless otherwise noted. For the temperature shift ($^{ts}$) experiment with larvae (Fig 3B), the temperature was shifted from 18°C to 30°C 48 h before dissection of L3 larvae. For clone induction in wing discs (Fig 3C), a heat shock at 37°C for 15 min was given to larvae 48 h before dissection of L3 larvae. The fly stocks used in this study are listed in Table S1.

### Immunofluorescence and imaging

Wing discs or adult midguts were immunostained as described previously (Ciesielski et al, 2022; Ikegawa et al, 2023). We used the following antibodies and fluorescent dyes at the indicated dilutions:
rabbit anti-phospho-H3 (1:1,000, 06-570; Merck).
mouse anti-armadillo (1:100, N2 7A1; DSHB).
mouse anti-Prospero (1:50, MR1A; DSHB).
rabbit anti-cleaved Dcp-1 (1:500, 9578; Cell Signaling).
rabbit anti-ß-galactosidase (1:300, 559761; MP Biomedicals).
mouse anti-HA (1:500, 901513; BAB).
DAPI (1:500, D9542; Sigma-Aldrich).
Alexa Fluor secondary antibodies (1:500, A11008, A11036, and A32723; Thermo Fisher Scientific).

Fluorescence images were acquired with a confocal microscope (LSM 780, 880, and 900; Zeiss) at 10x, 20x, and 40x magnification. Midgut pictures for quantification were taken in the R4a-b region.

### TUNEL assay

The TUNEL assay was performed using ApopTag Red In Situ Apoptosis Detection Kit (Millipore) as described previously (Ikegawa et al, 2023). Wandering third instar larvae or adult flies were

dissected in 1x PBS and fixed for 30 min and 1 h respectively, in 1x PBS with 4% PFA at RT. After fixation, samples were washed with PBS/0.1% Triton X-100 and incubated in equilibration buffer (ApopTag kit; Millipore) for 10 s. Then, samples were incubated in reaction buffer (TdT enzyme; ratio 7:3; ApopTag kit) at 37°C for 1 h. The TdT reaction mix was replaced with stop buffer (diluted 1:34 in dH$_2$O; ApopTag kit) and incubated for 10 min at RT. Samples were then washed with PBS/0.1% Triton X-100 three times and incubated with anti-digoxigenin antibody solution (diluted 31:34 in blocking solution; ApopTag kit) overnight at 4°C. Samples were then washed with PBS/0.1% Triton X-100 three times again and mounted.

### Quantification of GFP-positive cell density and GFP area

GFP-expressing cells were counted manually using Imaris 9.5.1, and Fiji was used to measure the tissue area. The GFP+ cell density was calculated by dividing the number of cells by tissue area. Fiji was used to measure the area of GFP-expressing cells in the wing disc.

### Quantification of AP-1-GFP intensity

Confocal images were acquired. DAPI signals or nls-RFP signals were used to allocate the nuclear area of the ISC, and GFP intensity was measured in the selected area using Fiji.

### Quantitative PCR with reverse transcription

Total RNA was extracted from 10 wing discs or five midguts per sample using Maxwell RSC simplyRNA Tissue Kit (Promega). 300 ng of extracted RNA was reverse-transcribed using Prime-Script RT Master Mix (RR036A; TaKaRa). Real-time PCR was performed using FastStart Essential DNA Green Master (Roche) with LightCycler 96 (Roche). Transcript levels were normalized with RpL32 in the same samples.

Oligonucleotide sequences used for qRT-PCR were as follows: RpL32 forward: 5′-CCAGCATACAGGCCCAAGATCGTG-3′; RpL32 reverse: 5′-TCTTGAATCCGGTGGGCAGCATG-3′; DIAP-1 forward: 5′-CCCAGTATCCC-GAATACGCC-3′; DIAP-1 reverse: 5′-TCTGTTTCAGGTTCCTCGGC-3′; and Rpr forward: 5′-ACGGGGAAAACCAATAGTCC-3′; Rpr reverse: 5′-TGGCTC TGTGTCCTTGACTG-3′.

### ATAC-seq analysis

To perform ATAC-seq, ISCs were isolated by FACS. Midguts from female *esg*-GFP flies (6 d old) were dissected in 1x PBS and dissociated in 1 mg/ml Elastase solution (Wako) for 1 h at RT. The dissociated cells were resuspended in 1xPBS after centrifuge, filtered through a 40-μm filter, and kept on ice until sorting. Cell sorting was performed using the SH800S cell sorter (Sony). Sorted cells were resuspended in 50% FBS/10% DMSO/40 *Drosophila* Schneider's medium after centrifugation and stored at –80°C. Two samples, A (50,000 cells) and B (100,000 cells), were sent to Active Motif. Sample preparation and sequencing analysis for ATAC-seq were performed by Active Motif.

### Starvation assay

For measuring the starvation resistance, 5- to 7-d-old virgin female flies were used and reared in each vial with ~20 flies. After maturation, flies were fed with the starvation food at 25°C. The starvation food contained 0% sucrose, 0% yeast, and 1.5% agar. To measure starvation resistance, we counted the number of dead flies every day.

### Generation of transgenic flies

For making transgenic flies, DNA for Rpr, Rpr-MTS, Rpr-Golgi, or Rpr-ER was synthesized by GenScript and cloned into the pUASTattB vector. For Rpr-MTS, the last 24 amino acids of the HID protein were added to the C-terminus of Rpr as previously described (Sandu et al, 2010). For Rpr-ER, a 17–amino acid calreticulin signal peptide was added to the N-terminus of Rpr along with a KDEL sequence at the C-terminal end. For Rpr-Golgi, the N-terminal 81 amino acids of human beta-1,4-galactosyltransferase (GalT) signal peptide were added to the N-terminus of Rpr. Transgenic flies were generated by inserting the plasmid into the attP2 on the third chromosome by BestGene.

### Statistical analysis

Statistical tests and sample sizes used are indicated in the figure captions. All the statistical analyses were conducted in GraphPad Prism. Sample sizes were determined empirically based on the observed effects. All experiments were conducted at least twice. Data shown in figures are cumulative data from multiple experiments or a representative dataset of several experiments.

## Data Availability

This study includes no data deposited in external repositories.

## Supplementary Information

## Acknowledgements

We thank Iswar Hariharan, Donald Fox, Hermann Steller, Yu-ichiro Nakajima, Takashi Nishimura, and Hyung Don Ryoo, and TRiP at Harvard Medical School, the Bloomington Stock Center, and the Zurich ORFeome project for fly stocks. We thank the Yoo laboratory members, Natsuki Shinoda and Takashi Nishimura, for helpful comments on the article. This work was supported by RIKEN Junior Research Associate Program to S Sulekh and AMED-PRIME (17939907), JSPS KAKENHI (JP16H06220), JSPS KAKENHI (JP22H02807), and JST FOREST (JPMJFR216F) to SK Yoo.

### Author Contributions

S Sulekh: conceptualization, formal analysis, investigation, visualization, methodology, and writing—review and editing.

Y Ikegawa: investigation, methodology, and writing—review and editing.

S Naito: investigation, methodology, and writing—review and editing.

A Oji: investigation, methodology, and writing—review and editing.

I Hiratani: methodology, project administration, and writing—review and editing.

SK Yoo: conceptualization, funding acquisition, investigation, project administration, and writing—original draft, review, and editing.

## Conflict of Interest Statement

The authors declare that they have no conflict of interest.

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
