## [Reviewer comments · Life Science Alliance]

Life Science Alliance

A feedback loop that drives cell death and proliferation and its defect in intestinal stem cells

Shivakshi Sulekh, Yuko Ikegawa, Saki Naito, Asami Oji, Ichiro Hiratani, and Sa Kan Yoo

DOI: <https://doi.org/10.26508/lsa.202302238>

Corresponding author(s): Sa Kan Yoo, RIKEN Center for Biosystems Dynamics Research

Review Timeline:

Submission Date:	2023-06-27
Editorial Decision:	2023-07-27
Revision Received:	2023-12-05
Editorial Decision:	2023-12-21
Revision Received:	2024-01-05
Accepted:	2024-01-05

Transaction Report:

July 27, 2023

Re: Life Science Alliance manuscript #LSA-2023-02238-T

Dr. Sa Kan Yoo
RIKEN
Kobe
Japan

Dear Dr. Yoo,

Thank you for submitting your manuscript entitled "A positive feedback circuit that simultaneously drives cell death and proliferation" to Life Science Alliance. The manuscript was assessed by expert reviewers, whose comments are appended to this letter. We invite you to submit a revised manuscript addressing the Reviewer comments.

Thank you for this interesting contribution to Life Science Alliance. We are looking forward to receiving your revised manuscript.

Sincerely,

B. MANUSCRIPT ORGANIZATION AND FORMATTING:

Reviewer #1 (Comments to the Authors (Required)):

Sulekh et al. report that a mutually positive feedback loop between JNK and apoptotic caspase activity can simultaneously drive cell death and cell proliferation depending on the cellular context. While JNK/caspase interaction leads to cell death in the wing discs as seen before, the outcome is cell proliferation in the adult Intestinal Stem Cells (reported here). Several conclusions are supported by the data and include the important finding that BH3 only protein Synr can induce cell proliferation in the intestine in a JNK and effector caspase-dependent manner and that effector caspase Dcp1 is needed for optimal JNK activation in this context. But the explanation that JNK promotes proliferation instead of death cell-autonomously because it cannot activate Rpr needs more supportive data. Specifically:

1. The authors state that Synr induces cell death in the wing disc but not in the intestine, but TUNEL staining is shown for the wing disc only. To support their conclusion, TUNEL staining should be shown for the intestine after Synr induction. Related, the authors propose that a positive feedback from JNK to caspases is defective in ISC/EB but do not actually assay for caspase cleavage/activation and cell death (TUNEL) after JNK induction in the intestine. This should be done.
2. In Fig. S1C, *esg>synr* increased not only GFP+ cells but also GFP- cells by about 1.5 folds (I count ~50 in the control vs. ~80 in *synr* panels). Since *synr* is expressed in GFP+ cells, this seems to be a non-autonomous effect. So, are all pH3+ cells GFP+? This should be checked for overexpression of *synr*, *Dcp1* and *hep-CA*. Distinguishing autonomous/non-autonomous effects is relevant because in Apoptosis-induced Proliferation (AiP) in the wing disc, JNK acts in cells with caspase activity to induce non-autonomous proliferation in the neighbors. AiP is stronger if cells activate caspases but do not die because mitogenic signaling is prolonged in this case. Can this be what is happening in the intestine? *Synr* may be a poor activator of caspases, prolonging AiP and inducing proliferation of the neighbors (both GFP+ and GFP-). This would be a different interpretation of the data than the cell-autonomous positive feedback that the authors propose.
3. Fig. 3. HA-rpr localization (or the lack of it) at the mitochondria is not convincing. The HA signal seems to be everywhere including where the mito-cherry signal is. How does HA-rpr-*mts*/mito-cherry co-localization look? Can this be contrasted with HA-rpr/mito-cherry localization? Regarding the ability of *rpr-*mts** but not *rpr* to affect cell number, are the expression levels comparable? Given that *rpr* on chromosome II can reduce cell number, I am not convinced that mito localization matters. The authors describe the effect of chromosome II *rpr* as 'mild', but it seems comparable to me to the effects of dark RNAi or DN-JNK in Fig. 2A.
4. Overexpression of *dcp1* only accounts for 26% of pH3 cells (~8 for *dcp1* vs ~30 for *synr*). Is *Drice* or *Dronc* playing a role in ISC proliferation? It is known that *Dronc* can activate JNK. Is it full length *Dcp1* or cleaved *Dcp1* that is overexpressed? If full length, is caspase activity not needed? Cleavage should be checked by antibody staining.

Presentation issues to address.

1. The number of biological replicates should be provided in the figure legends.
2. Scale bars are missing in Fig 3F.
3. Please describe the reagents better. I assume *rpr-*mts** is *rpr* with the mito localization signal, but this is not clearly stated, nor are changes made to *Rpr* to localize it to the mitochondria described. *rtts* is not described or listed in the stock list. What is the purpose of anti-Arm, pros staining?
4. Fig. 2G. Why is *Dcp1* activated only in about half of the nub domain with constitutive JNK activity?
5. Fig. 2I. Why is *Dcp1* activated only in a small fraction of nub expression domain where *synr* is presumably expressed throughout the domain? Again, *Synr* seems to be a poor activator of caspases, and this could be inducing AiP.
6. Based on the diagram in Fig. 1A, inhibiting *rpr*, *hid*, and *grim* should not affect proliferation induced by *Synr*. Does miRNA-RHG that is already used in Fig. 4 show an effect in this context?

Reviewer #2 (Comments to the Authors (Required)):

The manuscript by Sulekh et al. provides valuable insights into the mechanistic approach that explains the dual role of caspases in vivo. The study demonstrates a positive feedback loop between JNK and caspase, shedding light on how this loop promotes both cell death and proliferation. The authors also highlight that a failed JNK-caspase feedback loop not only suppresses cell death signals but also hinders proliferation. The study focuses on ISC cells and reveals that defects in this mechanism lead to a

tendency towards proliferation in ISCs.

Overall, this study significantly contributes to our understanding of how the JNK-caspase pathway regulates the balance between tissue-specific cell death and proliferation.

The results presented in the manuscript are convincing and represent a substantial advancement in the field. However, I have some minor suggestions for improvement:

1. In Figure 1C, the authors state that "Synr induces mitosis, which occurs only in ISCs among cells in the gut (Fig 1C)." To provide evidence for this claim, it would be helpful to include co-labelling staining for esg-GFP cells and anti-pH3 in the figure.
2. The phrase "Forced localization of Rpr to mitochondria depleted ISCs completely (Fig 3D-E) in the whole gut, except the R3 region" is referring only to Figure 3E. Please correct this statement accordingly. Additionally, if data regarding the absence of depletion of ISCs in the R3 region after forced mitochondrial localization of Rpr is not shown, consider mentioning it as "data not shown" or provide it as a supplementary figure/panel. It would also be valuable to discuss any possible explanations for the absence of ISC depletion in the R3 region after forced mitochondrial localization of Rpr.
3. Please transfer the following phrase from the discussion section to the results section: "We also found that Dcp1 activation initially induces proliferation in ISCs (Fig 1H), which is followed by cell death if Dcp-1 is expressed for a longer time (Fig S3A)." This will help present the findings in a more orderly manner.

Reviewer #3 (Comments to the Authors (Required)):

In this manuscript, the authors report a caspase-activating positive feedback circuit that promotes the proliferation of Intestinal Stem Cells (ISCs) instead of killing them. Specifically, they show that genetic conditions that initiate caspase-dependent apoptosis in other tissues paradoxically increase ISC numbers in the fly gut. They further show that blocking caspases or JNK signaling blocks this proliferation response. Unlike other tissues, JNK signaling does not activate caspases in ISCs for two reasons: The Diap1 inhibitor, reaper, is transcriptionally silenced. In addition, Reaper protein fails to localize to the mitochondrial membrane. Instead, JNK appears to act through Yki to promote cell proliferation.

Overall, the authors report an interesting observation through well-designed experiments. The data are accompanied by proper quantification and statistical analysis. I only have a few minor questions, which I hope the authors could clarify:

1. The authors use the term "positive feedback between JNK and caspases" throughout the manuscript (including the title and the abstract). But it appears that there is no such positive feedback acting in ISCs, as JNK cannot activate caspases in these cells (see authors figure 4B). If the authors wish to stick with this term, please provide evidence. For example, are there any small amounts of cleaved Dcp-1 generated by JNK activation in ISCs? Otherwise, I suggest avoiding the term "positive feedback loop" for ISCs.
2. The authors show that JNK activation does not cause apoptosis in ISCs, and focus their explanation around Reaper. However, Figure 1F, G shows that ISCs can tolerate active caspases at least for a few days (when activated by Synr overexpression). This means that the block in apoptosis lies downstream of Reaper. The authors might want to speculate how this occurs. If ISCs can resist active caspases that lie downstream of Reaper, it would mean that Reaper regulation in ISCs account for only a part of ISC's resistance to cell death.

We thank the editors and the reviewers for their helpful comments. We addressed almost all comments experimentally. Because of their insightful comments, now the manuscript improved tremendously. We hope that the editors and the reviewers would be satisfied with our revised manuscript. Following is a point-by-point response.

Reviewer #1:

Sulekh et al. report that a mutually positive feedback loop between JNK and apoptotic caspase activity can simultaneously drive cell death and cell proliferation depending on the cellular context. While JNK/caspase interaction leads to cell death in the wing discs as seen before, the outcome is cell proliferation in the adult Intestinal Stem Cells (reported here). Several conclusions are supported by the data and include the important finding that BH3 only protein Synr can induce cell proliferation in the intestine in a JNK and effector caspase-dependent manner and that effector caspase Dcp1 is needed for optimal JNK activation in this context. But the explanation that JNK promotes proliferation instead of death cell-autonomously because it cannot activate Rpr needs more supportive data. Specifically:

1. The authors state that Synr induces cell death in the wing disc but not in the intestine, but TUNEL staining is shown for the wing disc only. To support their conclusion, TUNEL staining should be shown for the intestine after Synr induction. Related, the authors propose that a positive feedback from JNK to caspases is defective in ISC/EB but do not actually assay for caspase cleavage/activation and cell death (TUNEL) after JNK induction in the intestine. This should be done.

We thank the reviewer for suggesting these experiments. We have now added a panel for TUNEL staining in the gut after sayonara induction in Fig 1C. For caspase activation and cell death by JNK, we have now included cleaved-dcp-1 staining and TUNEL staining in Fig 2D.

2. In Fig. S1C, *esg>synr* increased not only GFP+ cells but also GFP- cells by about 1.5 folds (I count ~50 in the control vs. ~80 in *synr* panels). Since *synr* is expressed in GFP+ cells, this seems to be a non-autonomous effect. So, are all pH3+ cells GFP+? This should be checked for overexpression of *synr*, *Dcp1* and *hep-CA*.

Many thanks for this comment. In the original manuscript, we failed to emphasize this enough, but in the *Drosophila* gut, only ISCs (GFP positive cells here) are mitotic. This has been well established and accepted in the field (Micchelli & Perrimon, 2006; Miguel-Aliaga *et al*, 2018; Ohlstein & Spradling, 2006; Sasaki *et al*, 2021). To clarify this point, we also added a new picture that shows all pH3 positive cells are GFP positive (Fig S1C). Additionally, the reviewer's observation is correct: more proliferation of ISCs leads to more numbers of differentiated cells, inducing a crowded condition in the gut. We added a text "Due to ISC proliferation by *synr*, the gut becomes more crowded with differentiated cells" in the figure legend (Fig S1D).

Distinguishing autonomous/non-autonomous effects is relevant because in Apoptosis-induced Proliferation (AiP) in the wing disc, JNK acts in cells with caspase activity to induce non-autonomous proliferation in the neighbors. AiP is stronger if cells activate caspases but do not die because mitogenic signaling is prolonged in this case. Can this be what is happening in the intestine? *Synr* may be a poor activator of caspases, prolonging AiP and inducing proliferation of the neighbors (both GFP+ and GFP-). This would be a different interpretation of the data than the cell-autonomous positive feedback that the authors propose.

This reviewer's comment is correct. In imaginal discs such as eye discs and wing discs JNK is known to induce non-cell autonomous proliferation. But, we also acknowledge that JNK can induce autonomous cell proliferation through Yki in imaginal discs (Enomoto *et al*, 2015; La Marca & Richardson, 2020; Sun & Irvine, 2011).

In the original manuscript, we concluded synr-induced ISC proliferation is a cell-autonomous event. In the revised manuscript, we put a more nuanced discussion that we cannot completely rule out a possibility that non-autonomous signals are involved. However, we are still more inclined to think our observation is cell autonomous.

Let us share our humble and frank opinion on difficulty to completely rule out non-autonomous mechanisms below.

In case of imaginal discs, when we make a decision whether events are cell autonomous or not, we often generate scattered clones and examine whether we see phenotypes or not in clones as a common practice. In that sense, ISCs are already sparsely located, which makes the situation the same with the clone experiments in imaginal discs. But, we acknowledge that, strictly speaking, both in imaginal discs and in ISCs, we cannot completely exclude the possibility of non-cell autonomous communication within clones or among clones. Thus, both in imaginal discs and ISCs, a more vigorous approach such as making twin spot clones should be necessary (but even with twin spots, we cannot exclude the possibility of non-autonomous communication within clones...). Due to technical reason in the gut, we are not making twin spot clones in the current manuscript. Thus, we acknowledge that we cannot rule out the non-autonomous possibility completely.

However, we'd like to note that in the gut, overwhelming literature supports that JNK induces cell autonomous proliferation in ISCs (Biteau *et al*, 2008; Biteau & Jasper, 2011; Buchon *et al*, 2009; Herrera & Bach, 2021; Hu & Jasper, 2019; Loudhaief *et al*, 2017; Meng & Biteau, 2015; Mundorf *et al*, 2019): JNK induces cell autonomous ISC proliferation through autonomous mechanisms involving Ras/Erk, Sox21a, Wdr62, Kif1a etc. Additionally, a recent paper made a small number of clones in the gut and found that JNK is important for clone proliferation (Quintero & Bangi, 2023) (Yet again, strictly speaking, this does not exclude a possibility of intra- or inter-clone communication). We also manipulated multiple genes cell autonomously (sayonara and DN-JNK in the same cells, for example). Furthermore, through preliminary experiments, we confirmed that inhibiting DPP or wingless, both of which are important for non-cell autonomous proliferation in imaginal discs, does not suppress synr-induced proliferation, although this needs further validation.

Taken together, as mentioned above, we are more inclined to suggest that synr-induced ISC proliferation is cell autonomous. We appreciate the reviewer's comment and acknowledge that we cannot completely rule out the possibility of non-autonomous event within/among clones, the same with the imaginal disc situation. We took into account all of this and changed our statement and included a text that we cannot completely exclude a possibility of non-autonomous mechanism in discussion.

Additionally, although this does not address the reviewer's comment directly, in the revised paper, we included data that demonstrate in the wing disc, rpr/synr induces JNK activation autonomously (Fig 3C).

3. Fig. 3. HA-rpr localization (or the lack of it) at the mitochondria is not convincing. The HA signal seems to be everywhere including where the mito-cherry signal is. How does HA-rpr-mts/mito-cherry co-localization look? Can this be contrasted with HA-rpr/mito-cherry localization? Regarding the ability of rpr-mts but not rpr to affect cell number, are the expression levels comparable? Given that rpr on chromosome II can reduce cell number, I am not convinced that mito localization matters. The authors describe the effect of chromosome II rpr as 'mild', but it seems comparable to me to the effects of dark RNAi or DN-JNK in Fig. 2A.

We agree with this comment that the HA-rpr localization picture does not have a high resolution. Originally, we expected to see better signals of HA-rpr in ISCs, but we found it difficult mainly due to the extremely small size of ISCs. To improve the image resolution, we spent more than a year by using different HA antibodies in vain. We also made new transgenics of rpr-flag, rpr-GFP, GFP-rpr but, to our disappointment, these tagging prevented function of rpr. But, we hope that the reviewer appreciates that our data demonstrate rpr-HA does not accumulate in mitochondria, rather, there are less signals of rpr-HA in the regions with the highest mitoCherry. To emphasize this, we inserted a magnified inset in the revised fig (Fig 5C).

Unfortunately, rpr-mts, which we obtained from Hermann Steller lab, does not have the HA tag, but they extensively performed biochemical and cell biological characterization of rpr-mts and rpr in the previous paper (Sandu *et al*, 2010). Since we thought our data of HA-rpr localization do not have the highest resolution, we used rpr-mts for the functional analysis.

Regarding comparison of rpr (2) and dark RNAi/DN-JNK, the former is just rpr expression and the latter is a combination with sayonara, which makes the comparison difficult.

To address the comment on expression levels, we newly constructed rpr with/without mitochondrial targeting using the 3xFlag tag and generated transgenics. To our devastation, the rpr-3xflag and rpr-mts-3xflag lost their protein function due to the flag tagging. So, we resorted to non-flagged Rpr. Again, we made rpr-WT and rpr-mts from scratch by ourselves and inserted them to the same landing site on the 3rd chromosome (attP 2) to prevent position effects on the transgene expression. We confirmed that rpr-mts was much more effective than Rpr-WT (Fig S3C). As additional controls for rpr-mts, we also constructed rpr with ER- or Golgi-targeting signals and found that only mitochondrial targeting leads to complete ablation of ISCs (Fig 5D).

4. Overexpression of dcp1 only accounts for 26% of pH3 cells (~8 for dcp1 vs ~30 for synr). Is Drice or Dronc playing a role in ISC proliferation? It is known that Dronc can activate JNK. Is it full length Dcp1 or cleaved Dcp1 that is overexpressed? If full length, is caspase activity not needed? Cleavage should be checked by antibody staining.

Regarding the relatively weak induction of proliferation by Dcp1, we reason that it is due to its strong induction of cell death (Fig S1F). Dcp1 expression is simply too strong, inducing apoptosis without requiring the feedback loop amplification.

To our surprise, our preliminary experiments suggested neither Drice nor Dronc is important for synr-induced ISC proliferation. We used previously published RNAs (Drice: BL32403 and v28064, Dronc: BL32963, v23033 and v23035). Rather, we found that among initiator caspases, Dredd was important for synr-induced ISC proliferation (right data). We are relatively confident about Dronc and Drice since we used multiple RNAs, but negative data with RNAs always require more careful validation, so we prefer not to include these data in the current manuscript.

[Figure removed by editorial staff per authors' request]

A full length Dcp-1 was used for overexpression, which gets cleaved and can be detected with cleaved-dcp-1 antibody (Fig S1F).

Presentation issues to address.

1. The number of biological replicates should be provided in the figure legends.

Thank you for the suggestion, now we included the number of replicates in figure legends.

2. Scale bars are missing in Fig 3F.

Thank you for noticing our careless mistake, we have now included the scale bar.

3. Please describe the reagents better. I assume rpr-mts is rpr with the mito localization signal, but this is not clearly stated, nor are changes made to Rpr to localize it to the mitochondria described. rmts is not described or listed in the stock list. What is the purpose of anti-Arm, pros staining?

We apologize for having missed citing the previous paper from Hermann Steller's lab. As mentioned above, Rpr and rpr-mts have been extensively and convincingly characterized in the previous paper (Sandu *et al.*, 2010). We cited this important paper in a more visible manner in this revision. We also apologize for missing citing the appropriate paper (Smith-Bolton *et al.*, 2009) for rmt^{ts}. We update the stock list included in the revised paper.

Regarding Armadillo/prospiero staining, in the gut filed, it is a common practice to stain ISCs and enteroendocrine cells using the two antibodies for them simultaneously (for example, Fig 1A of this paper (Biteau & Jasper, 2014)). Armadillo strongly labels plasma membranes of small cells such as ISCs and enteroendocrine cells while prospiero labels the nucleus of enteroendocrine cells. Although both armadillo and prospiero antibodies were generated in mice, the patterns are different, so the gut researchers often use them together to distinguish ISCs and enteroendocrine cells.

4. Fig. 2G. Why is Dcp1 activated only in about half of the nub domain with constitutive JNK activity?

First, based on our experience, we have an impression that the center of the pouch is generally more sensitive to cell death than the periphery of the pouch. Thus, whenever we induce apoptosis, we start to see caspase activation (cDcp1 and GC3Ai) in the center of the pouch (See the pic below). This observation might be related to the observation that in the middle of

the pouch there is more proliferation (Mao *et al*, 2013). It's also possible that basally extruded apoptotic cells tend to accumulate towards the center of the pouch.

But somehow with JNK, cell death is more dominantly observed in the ventral periphery of the pouch (But, please note that with GC3Ai, which is more sensitive than cDCP1, we clearly see GC3Ai signals in the center of the pouch too). Currently, we do not know why this occurs, but it is known that Eiger/JNK-induced cell death is not solely apoptotic (Igaki *et al*, 2002). JNK was previously demonstrated to induce some necrotic death (Li *et al*, 2019). Thus, we speculate that constitutively active Hep

is so strong and might be inducing necrotic cell death, which somehow might be occurring more in the ventral part of the pouch. We added the following text in the corresponding legend: “For unknown reason, JNK-mediated caspase activation is more pronounced in the ventral periphery of the wing pouch.”

5. Fig. 2I. Why is Dcp1 activated only in a small fraction of nub expression domain where synr is presumably expressed throughout the domain? Again, Synr seems to be a poor activator of caspases, and this could be inducing AiP.

Please see our response above.

6. Based on the diagram in Fig. 1A, inhibiting rpr, hid, and grim should not affect proliferation induced by Synr. Does miRNA-RHG that is already used in Fig. 4 show an effect in this context?

Previously we extensively characterized synr, which acts as a BH3-only protein upstream of the Bcl2 pathway (Ikegawa *et al*, 2023). Since Diap1 constitutively suppresses caspases, which rpr/hid/grim antagonizes, miRNA-RHG is expected to suppress Synr's effect. If we inhibit caspases by Diap1 induction (through miRNA-RHG), it should suppress Synr-mediated caspase activation too. We included this data (Fig S1E).

Reviewer #2 (Comments to the Authors (Required)):

The manuscript by Sulekh et al. provides valuable insights into the mechanistic approach that explains the dual role of caspases in vivo. The study demonstrates a positive feedback loop between JNK and caspase, shedding light on how this loop promotes both cell death and proliferation. The authors also highlight that a failed JNK-caspase feedback loop not only suppresses cell death signals but also hinders proliferation. The study focuses on ISC cells and reveals that defects in this mechanism lead to a tendency towards proliferation in ISCs.

Overall, this study significantly contributes to our understanding of how the JNK-caspase pathway regulates the balance between tissue-specific cell death and proliferation. The results presented in the manuscript are convincing and represent a substantial advancement in the field. However, I have some minor suggestions for improvement:

We appreciate this reviewer's enthusiasm with our paper.

1. In Figure 1C, the authors state that "Synr induces mitosis, which occurs only in ISCs among cells in the gut (Fig 1C)." To provide evidence for this claim, it would be helpful to include co-labelling staining for esg-GFP cells and anti-pH3 in the figure.

We thank the reviewer for the suggestion, we have now added a co-labeled image of mitotic ISCs in Fig S1C. Please also see our comment to the reviewer 1.

2. The phrase "Forced localization of Rpr to mitochondria depleted ISCs completely (Fig 3D-E) in the whole gut, except the R3 region" is referring only to Figure 3E. Please correct this statement accordingly. Additionally, if data regarding the absence of depletion of ISCs in the R3 region after forced mitochondrial localization of Rpr is not shown, consider mentioning it as "data not shown" or provide it as a supplementary figure/panel. It would also be valuable to discuss any possible explanations for the absence of ISC depletion in the R3 region after forced mitochondrial localization of Rpr.

We appreciate the reviewer's keen eyes for spotting the mistake, we have made the appropriate edits. We have now included the data of the R3 region in Fig S3A.

Currently, we do not know why ISCs at R3 are more resistant to apoptosis. R3 is unique in a sense that its environment shows a very low PH, which is a harsh condition for cells. We speculate that since ISCs at R3 need to survive the low pH harsh environment, they might be especially resistant to cell death. We inserted this discussion in text of the revised manuscript.

3. Please transfer the following phrase from the discussion section to the results section: "We also found that Dcp1 activation initially induces proliferation in ISCs (Fig 1H), which is followed by cell death if Dcp-1 is expressed for a longer time (Fig S3A)." This will help present the findings in a more orderly manner.

We appreciate the suggestion and moved the time series of Dcp-1 expression to Fig S1F.

Reviewer #3 (Comments to the Authors (Required)):

In this manuscript, the authors report a caspase-activating positive feedback circuit that promotes the proliferation of Intestinal Stem Cells (ISCs) instead of killing them. Specifically, they show that genetic conditions that initiate caspase-dependent apoptosis in other tissues paradoxically increase ISC numbers in the fly gut. They further show that blocking caspases

or JNK signaling blocks this proliferation response. Unlike other tissues, JNK signaling does not activate caspases in ISCs for two reasons: The Diap1 inhibitor, reaper, is transcriptionally silenced. In addition, Reaper protein fails to localize to the mitochondrial membrane. Instead, JNK appears to act through Yki to promote cell proliferation.

Overall, the authors report an interesting observation through well-designed experiments. The data are accompanied by proper quantification and statistical analysis. I only have a few minor questions, which I hope the authors could clarify:

1. The authors use the term "positive feedback between JNK and caspases" throughout the manuscript (including the title and the abstract). But it appears that there is no such positive feedback acting in ISCs, as JNK cannot activate caspases in these cells (see authors figure 4B). If the authors wish to stick with this term, please provide evidence. For example, are there any small amounts of cleaved Dcp-1 generated by JNK activation in ISCs? Otherwise, I suggest avoiding the term "positive feedback loop" for ISCs.

We thank the reviewer for noticing this. In fact, we also considered what the reviewer pointed out in the original manuscript, and that is why we didn't include "ISCs" in the original title. Here in this paper what we want to propose are two: 1) we found a positive feedback loop that simultaneously drives cell death and proliferation as a mechanistic framework. We are proposing this as a general conceptual framework. 2) the loop is defective in ISCs. Maybe in the original title and abstract, the second aspect was not emphasized enough. Thus, we changed our title to the following one: *A feedback loop that drives cell death and proliferation and its defect in intestinal stem cells*. We hope the reviewer would be satisfied with this arrangement.

2. The authors show that JNK activation does not cause apoptosis in ISCs, and focus their explanation around Reaper. However, Figure 1F, G shows that ISCs can tolerate active caspases at least for a few days (when activated by Synr overexpression). This means that the block in apoptosis lies downstream of Reaper. The authors might want to speculate how this occurs. If ISCs can resist active caspases that lie downstream of Reaper, it would mean that Reaper regulation in ISCs account for only a part of ISC's resistance to cell death.

Synr is a relatively weak caspase activator. In this setting, to fully activate caspase, we are proposing the feedback loop between caspase and JNK is necessary. In fact, this is consistent with the previous report by the Morata lab. They also proposed that the feedback loop between caspases and JNK is necessary for full activation of caspases to induce cell death (Shlevkov & Morata, 2012). The difference between their model and our model is that we are proposing that this feedback loop induces not only cell death but also proliferation signals. In contrast to the situation with Synr (weak caspase inducer), we found that a short-term expression of Dcp1 itself induces ISC proliferation (Fig 1I) in the beginning, but prolonged expression of Dcp1 kills ISCs (Fig S1F). In case of Dcp1 overexpression, it can kill ISCs simply because it is so strong that it does not require the feedback loop for its full activation. We believe our feedback model and its defect in ISCs fully explains our observation. We hope this explanation is reasonable.

Biteau B, Hochmuth CE, Jasper H (2008) JNK activity in somatic stem cells causes loss of tissue homeostasis in the aging *Drosophila* gut. *Cell Stem Cell* 3: 442-455

Biteau B, Jasper H (2011) EGF signaling regulates the proliferation of intestinal stem cells in *Drosophila*. *Development* 138: 1045-1055

Biteau B, Jasper H (2014) Slit/Robo signaling regulates cell fate decisions in the intestinal stem cell lineage of *Drosophila*. *Cell Rep* 7: 1867-1875

Buchon N, Broderick NA, Chakrabarti S, Lemaitre B (2009) Invasive and indigenous microbiota impact intestinal stem cell activity through multiple pathways in *Drosophila*. *Genes Dev* 23: 2333-2344

Enomoto M, Kizawa D, Ohsawa S, Igaki T (2015) JNK signaling is converted from anti- to pro-tumor pathway by Ras-mediated switch of Warts activity. *Dev Biol* 403: 162-171

Herrera SC, Bach EA (2021) The Emerging Roles of JNK Signaling in *Drosophila* Stem Cell Homeostasis. *Int J Mol Sci* 22

Hu DJ, Jasper H (2019) Control of Intestinal Cell Fate by Dynamic Mitotic Spindle Repositioning Influences Epithelial Homeostasis and Longevity. *Cell Rep* 28: 2807-2823 e2805

Igaki T, Kanda H, Yamamoto-Goto Y, Kanuka H, Kuranaga E, Aigaki T, Miura M (2002) Eiger, a TNF superfamily ligand that triggers the *Drosophila* JNK pathway. *EMBO J* 21: 3009-3018

Ikegawa Y, Combet C, Groussin M, Navratil V, Safar-Remali S, Shiota T, Aouacheria A, Yoo SK (2023) Evidence for existence of an apoptosis-inducing BH3-only protein, sayonara, in *Drosophila*. *EMBO J* 42: e110454

La Marca JE, Richardson HE (2020) Two-Faced: Roles of JNK Signalling During Tumourigenesis in the *Drosophila* Model. *Front Cell Dev Biol* 8: 42

Li M, Sun S, Priest J, Bi X, Fan Y (2019) Characterization of TNF-induced cell death in *Drosophila* reveals caspase- and JNK-dependent necrosis and its role in tumor suppression. *Cell Death Dis* 10: 613

Loudhaief R, Brun-Barale A, Benguetat O, Nawrot-Esposito MP, Pauron D, Amichot M, Gallet A (2017) Apoptosis restores cellular density by eliminating a physiologically or genetically induced excess of enterocytes in the *Drosophila* midgut. *Development* 144: 808-819

Mao Y, Tournier AL, Hoppe A, Kester L, Thompson BJ, Tapon N (2013) Differential proliferation rates generate patterns of mechanical tension that orient tissue growth. *EMBO J* 32: 2790-2803

Meng FW, Biteau B (2015) A Sox Transcription Factor Is a Critical Regulator of Adult Stem Cell Proliferation in the *Drosophila* Intestine. *Cell Rep* 13: 906-914

Micchelli CA, Perrimon N (2006) Evidence that stem cells reside in the adult *Drosophila* midgut epithelium. *Nature* 439: 475-479

Miguel-Aliaga I, Jasper H, Lemaitre B (2018) Anatomy and Physiology of the Digestive Tract of *Drosophila melanogaster*. *Genetics* 210: 357-396

Mundorf J, Donohoe CD, McClure CD, Southall TD, Uhlirva M (2019) Ets21c Governs Tissue Renewal, Stress Tolerance, and Aging in the *Drosophila* Intestine. *Cell Rep* 27: 3019-3033 e3015

Ohlstein B, Spradling A (2006) The adult *Drosophila* posterior midgut is maintained by pluripotent stem cells. *Nature* 439: 470-474

Quintero M, Bangi E (2023) Disruptions in cell fate decisions and transformed enteroendocrine cells drive intestinal tumorigenesis in *Drosophila*. *Cell Rep* 42: 113370

Sandu C, Ryoo HD, Steller H (2010) *Drosophila* IAP antagonists form multimeric complexes to promote cell death. *J Cell Biol* 190: 1039-1052

Sasaki A, Nishimura T, Takano T, Naito S, Yoo SK (2021) white regulates proliferative homeostasis of intestinal stem cells during ageing in *Drosophila*. *Nat Metab* 3: 546-557

Shlevkov E, Morata G (2012) A dp53/JNK-dependant feedback amplification loop is essential for the apoptotic response to stress in *Drosophila*. *Cell Death Differ* 19: 451-460

Smith-Bolton RK, Worley MI, Kanda H, Hariharan IK (2009) Regenerative growth in *Drosophila* imaginal discs is regulated by Wingless and Myc. *Dev Cell* 16: 797-809

Sun G, Irvine KD (2011) Regulation of Hippo signaling by Jun kinase signaling during compensatory cell proliferation and regeneration, and in neoplastic tumors. *Dev Biol* 350: 139-151

December 21, 2023

RE: Life Science Alliance Manuscript #LSA-2023-02238-TR

Dr. Sa Kan Yoo
RIKEN Center for Biosystems Dynamics Research
N404 RIKEN CDB Bldg. A 2-2-3 Minatojima-minamimachi, Chuo-ku
Kobe 650-0047
Japan

Dear Dr. Yoo,

Thank you for submitting your revised manuscript entitled "A feedback loop that drives cell death and proliferation and its defect in intestinal stem cells". We would be happy to publish your paper in Life Science Alliance pending final revisions necessary to meet our formatting guidelines.

- please address Reviewer 1's remaining comments
- please consult our manuscript preparation guidelines <https://www.life-science-alliance.org/manuscript-prep> and make sure your manuscript sections are in the correct order
- please add your main, supplementary figure, and table legends to the main manuscript text after the references section
- Figure S2 has only one panel, so there is no need to label it as A. Please correct the figure and its legend and callout in the manuscript text
- please add an Author Contributions section to your main manuscript text
- please label the References section

A. FINAL FILES:

B. MANUSCRIPT ORGANIZATION AND FORMATTING:

Sincerely,

Reviewer #1 (Comments to the Authors (Required)):

Sulekh et al. report that a mutually positive feedback loop between JNK and apoptotic caspase activity can simultaneously drive cell death and cell proliferation depending on the cellular context. While JNK/caspase interaction leads to cell death in the wing discs as seen before, the outcome is cell proliferation in the adult Intestinal Stem Cells (reported here). Several conclusions are supported by the data and include the important findings that BH3 only protein Synr can induce cell proliferation in the intestine in a JNK and effector caspase-dependent manner, that effector caspase Dcp1 is needed for optimal JNK activation in this context, and that JNK promotes proliferation instead of death cell-autonomously because it cannot activate Rpr. These add significantly to our current knowledge about the non-apoptotic roles of apoptotic caspases.

In the revised version, the authors have addressed all my concerns with one exception. I asked for the number of biological replicates, but the authors provided only the sample size. For example, in Fig. 1E, $n=15$ for each sample but are these 15 samples from a single experiment (one biological replicate) or from two different experiments (two biological replicates), or more? The key question is whether statistically significant differences observed in one experiment reproducible to the next experiment. Currently, information needed to distinguish the number of technical replicates from the number of biological replicates is missing.

Reviewer #2 (Comments to the Authors (Required)):

Following a comprehensive review of the revised manuscript, it is evident that the authors have diligently addressed all previous critiques, enhancing the clarity and rigour of their work. Consequently, I recommend the acceptance of the paper for publication.

Reviewer #3 (Comments to the Authors (Required)):

The revised manuscript fully addresses the points that I had raised in the original review.

January 5, 2024

RE: Life Science Alliance Manuscript #LSA-2023-02238-TRR

Dr. Sa Kan Yoo
RIKEN Center for Biosystems Dynamics Research
N404 RIKEN CDB Bldg. A 2-2-3 Minatojima-minamimachi, Chuo-ku
Kobe 650-0047
Japan

Dear Dr. Yoo,

Thank you for submitting your Research Article entitled "A feedback loop that drives cell death and proliferation and its defect in intestinal stem cells". It is a pleasure to let you know that your manuscript is now accepted for publication in Life Science Alliance. Congratulations on this interesting work.

DISTRIBUTION OF MATERIALS:

Again, congratulations on a very nice paper. I hope you found the review process to be constructive and are pleased with how the manuscript was handled editorially. We look forward to future exciting submissions from your lab.

Sincerely,
